# Recent Progress in Polyolefin Plastic: Polyethylene and Polypropylene Transformation and Depolymerization Techniques

**DOI:** 10.3390/molecules30010087

**Published:** 2024-12-29

**Authors:** Acácio Silva de Souza, Patricia Garcia Ferreira, Iva Souza de Jesus, Rafael Portugal Rizzo Franco de Oliveira, Alcione Silva de Carvalho, Debora Omena Futuro, Vitor Francisco Ferreira

**Affiliations:** Programa de Pós-Graduação em Ciências Aplicadas a Produtos para a Saúde, Laboratório de Inovação em Química e Tecnologia Farmacêutica, Faculdade de Farmácia, Universidade Federal Fluminense, Rua Doutor Mario Vianna, 523, Santa Rosa, Niterói 24241-000, RJ, Brazil; patricia.pharma@yahoo.com.br (P.G.F.); ivasouza.quimica@gmail.com (I.S.d.J.); rrizzo@id.uff.br (R.P.R.F.d.O.); alcionecarvalho@id.uff.br (A.S.d.C.); dfuturo@id.uff.br (D.O.F.)

**Keywords:** polymers, microplastics, pyrolysis, hydrogenolysis, sustainability

## Abstract

This paper highlights the complexity and urgency of addressing plastic pollution, drawing attention to the environmental challenges posed by improperly discarded plastics. Petroleum-based plastic polymers, with their remarkable range of physical properties, have revolutionized industries worldwide. Their versatility—from flexible to rigid and hydrophilic to hydrophobic—has fueled an ever-growing demand. However, their versatility has also contributed to a massive global waste problem as plastics pervade virtually every ecosystem, from the depths of oceans to the most remote terrestrial landscapes. Plastic pollution manifests not just as visible waste—such as fishing nets, bottles, and garbage bags—but also as microplastics, infiltrating food chains and freshwater sources. This crisis is exacerbated by the unsustainable linear model of plastic production and consumption, which prioritizes convenience over long-term environmental health. The mismanagement of plastic waste not only pollutes ecosystems but also releases greenhouse gases like carbon dioxide during degradation and incineration, thereby complicating efforts to achieve global climate and sustainability goals. Given that mechanical recycling only addresses a fraction of macroplastics, innovative approaches are needed to improve this process. Methods like pyrolysis and hydrogenolysis offer promising solutions by enabling the chemical transformation and depolymerization of plastics into reusable materials or valuable chemical feedstocks. These advanced recycling methods can support a circular economy by reducing waste and creating high-value products. In this article, the focus on pyrolysis and hydrogenolysis underscores the need to move beyond traditional recycling. These methods exemplify the potential for science and technology to mitigate plastic pollution while aligning with sustainability objectives. Recent advances in the pyrolysis and hydrogenolysis of polyolefins focus on their potential for advanced recycling, breaking down plastics at a molecular level to create feedstocks for new products or fuels. Pyrolysis produces pyrolysis oil and syngas, with applications in renewable energy and chemicals. However, some challenges of this process include scalability, feedstock variety, and standardization, as well as environmental concerns about emissions. Companies like Shell and ExxonMobil are investing heavily to overcome these barriers and improve recycling efficiencies. By leveraging these transformative strategies, we can reimagine the lifecycle of plastics and address one of the most pressing environmental challenges of our time. This review updates the knowledge of the fields of pyrolysis and hydrogenolysis of plastics derived from polyolefins based on the most recent works available in the literature, highlighting the techniques used, the types of products obtained, and the highest yields.

## 1. Introduction

Plastics have been the great solution humanity found for more efficient materials. In nature, there is no solution that creates a problem, as they do not survive natural selection.

Macroplastic waste continues to pose a significant challenge as it accumulates in the environment at alarming rates. While the benefits of plastics, such as their durability, versatility, and cost-effectiveness, are undeniable when they are properly utilized and disposed of, they can have great negative impacts on the environment. Recycling rates for macroplastics remain dismally low, averaging below 9%, far less than the 30% target needed to establish a truly circular economy. This discrepancy underscores a growing public dissatisfaction and calls for urgent action to improve collection, recycling, and waste management systems.

Scientists and innovators are actively seeking ways to transform this waste into a valuable resource. By reusing macroplastics as raw materials for the production of high-value chemical compounds, they aim to reduce environmental harm while supporting the materials and fuel industries. Chemical and mechanical recycling technologies have emerged as pivotal solutions in this endeavor. These industrial processes are increasingly being recognized for their potential to drive cost-effective and sustainable waste management practices. Whatever the method of reusing the large amount of macroplastics scattered around the world, there is growing interest in recycling plastics or transforming them into chemical products that can be used in other reactions, thus representing high-added-value waste [1]. From discarded waste, mechanically recycled and remanufactured plastics become new polymers for reuse, but mechanical recycling results in lower-quality materials.

Worldwide, more than 330 million tons of plastic are produced annually, with a considerable increase in production during the COVID-19 pandemic period, from 2019 to 2021. It is estimated that there are already 4.9 billion tons of plastic waste of varying sizes and chemical compositions, ubiquitous to all natural habitats, and those materials are spread in terrestrial and aquatic ecosystems. Projections indicate that in 2050, this amount will increase by 12 billion metric tons [2].

In 2019, the IUPAC began to envision the future of chemistry and launched the series entitled “Top Ten Emerging Technologies in Chemistry”, as part of an effort to widely promote the essential value of the chemical sciences and related fields, seeking to identify discoveries with the potential to transform the world. From 2020 to 2023 [3,4], various suggestions and opinions were provided, presenting several ideas for the development of new technologies capable of significantly impacting our society. In 2023, for the first time, the IUPAC selected two topics related to plastic recycling, namely, “Biological recycling of PET” and “Depolymerization of plastic waste”, to produce reusable building blocks, which include monomers, oligomers, or other value-added chemicals [5].

Many polymers are used in manufacturing plastics, with the most commonly used ones being mainly found in production parts and utensils (Figure 1). Monomers, the basic building blocks of polymers, are transformed into their final commercial form (e.g., packaging, furniture, parts, toys, household utensils, disposables, etc.) after being grouped together and processed, usually using heat. The most used polymers are polyethylene (PE) (**1**), polypropylene (PP) (**2**), polystyrene (PS) (**3**), and polyethylene terephthalate (PET) (**6**).

## 2. Macroplastics Are Not Waste

Discarded macroplastics should not be considered waste, but rather valuable resources for new uses, such as the production of fuels, alcohols, aldehydes, surfactants, and detergents, among other materials [6]. This cycle of reuse is called the circular economy of plastics, and it generates high added value for these materials, reducing the consumption of oil derivatives and the environmental impacts of this industry through reuse, recycling, and chemical transformation into other products. It is important to note that not all plastics can be recycled and, therefore, end up losing economic value while awaiting the development of technologies for their recovery [7]. The recovery and utilization of macroplastics involve initial stages such as collection, separation, processing, and marketing, aiming to prevent these materials from ending up in landfills, rivers, lakes, and oceans. These steps are essential parts of a city’s waste management by both the public sector and civil society and are crucial for the success of any project aimed at recycling these materials that pollute the environment. Figure 2 schematically shows the rates of recycling, dumping, and incineration of plastic waste consumed globally [8].

An alternative to reducing the amount of macroplastics is to transform them into other valuable goods and services. For example, combustion can be used to produce thermal or electrical energy (‘waste to energy’) [9]. Macroplastics are derived from oil and are, therefore, a valuable energy source compared to coal or other fossil fuels. In reality, much of the plastic discarded after use is incinerated in many countries to produce thermal or electrical energy using cement kilns or fluidized bed furnaces [10]. However, these processes contribute to an increase in atmospheric CO_2_ and particulate matter in the air as the plastic waste stream is mixed and heavily contaminated with other materials, including metals [11]. In addition, the combustion of plastics overlooks the fact that polymers could serve as raw materials for the production of other chemical materials with properties different from the original ones [12].

To improve the future of the planet, we need to shift from a linear economy to a circular economy for plastics with zero waste, in other words, a carbon-neutral economy. The circular economy involves continuously reusing plastics, with various strategies for recovering macroplastics (Figure 3).

Recycling, re-extrusion, or mechanical recycling is a type of primary recycling that uses discarded macroplastics (mainly poly-α-olefins) [13,14]. This process stimulates the circular economy by keeping the polymers within their original carbon chain, allowing them to be reused in the manufacture of other products and reducing the pressure on oil production [15,16].

Chemical recycling, on the other hand, is appealing because it generates higher value-added products through various types of chemical conversions. Currently, chemical recycling is mostly limited to condensation polymers and requires large volumes of plastics to be profitable. Among the applicable methods, depolymerization, a process that was developed a few years ago, is gaining increasing prominence in the utilization of macroplastics [17], with the search for new procedures and new catalysts being expanded due to the crisis of environmental pollution by macro- and microplastics.

There are three alternatives for the chemical transformation or cracking of polyolefins: gasification, pyrolysis, and hydrogenolysis. Gasification is a thermochemical process that converts plastic waste into a mixture of gases, primarily syngas (a blend of carbon monoxide (CO) and hydrogen (H_2_)), by heating the plastics at high temperatures (typically 700–1200 °C) in the presence of a controlled amount of oxygen, air, or steam. Unlike combustion, which results in complete oxidation, gasification is a partial oxidation process aimed at producing valuable gases for energy or chemical feedstocks. Pyrolysis and hydrogenolysis approaches promote the cleavage of polymeric chains and generate heterogeneous mixtures of polyolefins; they are carried out in reactors under pressure and heating in an inert atmosphere using metal catalysts. Pyrolysis and hydrogenolysis have been the most extensively researched technologies in recent years, thus being the focus of our study.

Catalytic processes used in oil cracking and lignin breakdown can be extended to polyolefin depolymerization, including alkane dehydrogenation/aromatization, transfer hydrogenation, and hydrogen cogeneration, as well as opportunities to use the polymer itself as a hydrogen source [18]. Figure 4 summarizes what is expected from the pyrolysis and hydrogenolysis processes. These aspects will be discussed below.

This article focuses exclusively on chemical upgrading processes, specifically chemolysis by pyrolysis and hydrogenolysis. It does not address the biochemical degradation of plastics or the production of biodegradable polymers. It is important to note that chemolysis refers to the chemical breakdown of polymers, a process commonly used to recycle plastics into their monomers or other valuable compounds. To ensure the efficiency of chemolysis and the production of high-quality outputs, waste pretreatment is essential. This involves steps such as shredding, washing, and separating contaminants.

## 3. Pyrolysis Cracking

Pyrolysis is a thermochemical decomposition process that involves breaking down organic material at high temperatures (300–900 °C) in the absence of oxygen, with or without a catalyst, making it suitable for many waste products, including highly complex plastics. The removal of oxygen is crucial because its presence can lead to combustion rather than decomposition. Thermal pyrolysis involves simply heating plastics to induce the cleavage of their bonds and produce shorter-chain hydrocarbons, while catalytic pyrolysis is carried out with a catalyst to lower the temperature and reaction time. In general, pyrolysis offers the advantage of high degradation efficiency but has the disadvantages of high energy consumption, low product selectivity, and difficulty in studying the degradation mechanism. Both thermal and catalytic pyrolysis have been studied in various reactors (batch, semi-batch, fixed bed, fluidized bed, batch with fixed bed, rotary, etc.), using different materials that are important for the efficiency of plastic depolymerization processes [19].

Pyrolysis represents a promising technology for converting waste into renewable energy, aligning with sustainability objectives and the circular economy. It is being studied more extensively than hydrogenolysis due to its cost-effectiveness and practicality. The pyrolysis process is already used to convert organic biomass and various waste materials that are considered useless (such as agricultural waste, wood, plastics, and municipal solid waste) into biochar, bio-oil, and synthesis gas (‘syngas’), thereby reducing the use of landfills and environmental pollution. Bio-oil can be refined to produce chemicals and other materials traditionally obtained from oil. However, it should be noted that pyrolysis has low efficiency for PVC [20]. The products obtained from the depolymerization of polyolefins, for example, vary according to the cracking conditions and the catalysts used (e.g., metal oxides, sulfated metal oxides, nanostructured zeolites, molecular sieves, metal carbonates, mesoporous materials, heterogeneous acids, zeolites, alumina, silica, etc.) [21]. The random splitting of C-C bonds into radicals generates complex mixtures of olefinic and cross-linked compounds.

In general, the pyrolysis depolymerization process produces three different phases, the proportion of which depends on the catalyst: a solid phase (coal or coke: 5–25% by weight), a liquid phase (tar, cycloparaffins, oligomers, and aromatics: 10–45% by weight), and a gas phase (volatile alkanes and alkenes), all of which are used for various purposes. A problem in the recycling or chemical transformation of polymers is the contaminants (organic, inorganic, paper, halogens, and metals) and various types of additives added to the plastics [22].

Polypropylene is very similar to polyethylene in that its structure only has C-H and C-C bonds. It is the second most used and discarded polymer, and therefore, the valorization of this material is extremely important to mitigate the presence of plastics in the environment.

The concept of recycling and upgrading polyethylene, polypropylene, and other polymers into oils with low molecular masses and functional carbon chains has been studied by various research groups worldwide, yielding promising results for a future where plastics are valuable starting materials [23,24,25,26,27]. This idea has become increasingly urgent as polyethylene waste accounts for approximately 30% of plastic waste. It is crucial to transform this waste into fuel oil, which is rich in alkanes, through the catalytic hydrogenolysis of the C-C bond.

Thermocatalytic depolymerization of polyolefins is one of the most economically promising strategies for creating higher-value-added products from plastic waste. As the chemical structures of polymers are very stable and have strong C-C bonds, this process requires a lot of thermal energy, which is usually provided by pyrolysis above 300 °C under the influence of a catalyst and in the absence of air, to depolymerize them. Under these conditions, the polymers are fragmented into simpler units and this process can be applied to industrial and household plastic waste, transforming the polyolefins into gases, liquids, and carbonized solid waste. The quality of the oil depends on the mixture of plastics, catalysts, and the conditions used in the pyrolysis [28]. The oily liquids resulting from pyrolysis consist mainly of hydrocarbons in the diesel boiling point range (180–380 °C), amounting to approximately 50–55% by volume. However, these crude liquids are not suitable for use as diesel fuel, but if distilled in the diesel boiling point range, they can be used in blends with conventional automotive diesel. In this way, plastics can be reused in another form and will no longer accumulate in the environment [29].

Ahmad et al. [30] obtained high selectivity for the pyrolysis of high-density polyethylene (HDPE) in the liquid fraction using nanostructured BaTiO_3_ doped with Pb at 350 °C to provide alkanes (73.4%), olefins (22.5%), and naphthalene (4.1%). There are many other catalysts for hydrocracking polystyrene with high yields of oils, liquids, and alkanes that can be used as lubricants and paraffin waxes or further processed into detergents and cosmetics [31]. Abbas-Abadi et al. [32] studied continuous pyrolysis on a pilot scale of different raw materials at temperatures between 430 and 490 °C and pressures between 0.1 and 2.0 bar. At the lowest pressure, the yield of low-density polyethylene pyrolysis oil reached 95% conversion by weight and its composition was α-olefins (37–42%) and n-paraffins (32–35%). With polypropylene, it was possible to form 84–91% iso-olefins (C9 and C15) and diolefins from oil. As for macroplastic waste, the pyrolysis oil yields were much lower and there was more char formation with metallic contamination. Whajah et al. [33] depolymerized polyethylene with high yields, induced by heating in the absence of hydrogen, at atmospheric pressure and with bifunctional catalysts based on zeolite containing dispersed Pt or Pt-Sn, obtaining hydrocarbons of up to 95% by weight after 2 h. The temperature of this process was 375 °C, and the distribution of products ranged from light gas to hydrocarbons in the range of gasoline and diesel. Hafeez et al. [34] carried out the pyrolysis of plastics at a temperature range of 600–700 °C in a fluidized bed reactor on a Pt/Al_2_O_3_ catalyst, obtaining oils and waxes after 6–8 h. Conversion during hydrotreatment reduced the reactivity of the pyrolysis oil and promoted the production of diesel and kerosene.

Wang et al. [35] studied catalytic thermolysis associated with sunlight to transform a mixture of plastic waste using an abundant Ni-based catalyst. Solar energy can be an important source of radiation to produce sustainable and efficient plastic pyrolysis. The process used a mixture of plastic waste, containing five types of polyolefins, polyester, and polyvinyl chloride, converting them into methane with a carbon yield of 98% and HCl with a chlorine yield of 91%.

The catalytic pyrolysis process has thermodynamic limitations that hinder the adsorption of polymers to the catalysts to promote the cracking of the chains. Kang et al. [36] tested catalytic reactions taking place inside mesoporous channels impregnated with Ru nanoparticles, allowing them to achieve an entropically more favorable stable transition state. This approach involves the synthesis of ruthenium catalysts distributed within mesoporous silica channels (SBA-15 < 150 μm particle size, pore size 10 nm, hexagonal pore morphology). The p-Ru/SBA-15 catalyst design resulted in improved catalyst performance in the conversion of polyethylene into high-value liquid fuels, especially diesel. When the Ru/SiO_2_ and Ru/C catalysts were evaluated, the mesoporous silica catalyst proved to be far superior in terms of the solid conversion rate, providing a great opportunity for the chemical recycling of plastic waste.

A very interesting strategy is the use of sacrificial solvents and co-fired hydrocarbons to cleave C-C bonds without the need to use hydrogen. In this field, Hancock and Rorrer carried out a comprehensive review of the low-temperature depolymerization of polyolefins in the absence of hydrogen and the tandem dehydrogenation and cross-metathesis of olefins for the depolymerization of polyethylene [37].

Cracking polyolefins in the presence of solvents provides benefits such as easier feeding of the reactants into the reactor and better heat and mass transfer. The presence of a hydrogen-donating solvent can improve the useful life of the catalyst and reduce the formation of coke residues. The catalytic cracking of polymers dissolved in solvents also allows higher value-added products to be obtained (e.g., tetralin, decalin, and methylcyclohexane) [38].

Wang et al. [39] developed an innovative strategy for the chemical conversion of polyethylene (**1**) via four catalytic reactions: tandem dehydrogenation to introduce an unsaturation into the saturated polyethylene chain, ethenolysis (metathesis reaction [40] with ethene, forming two terminal olefins), isomerization of the terminal olefin, and new ethenolysis (new metathesis reaction with ethene, forming propylene). In this method, which is detailed in Figure 1, a large excess of ethylene is needed to achieve the ethenolysis events per polymer chain and shift the reaction equilibrium towards propylene. This recent experiment demonstrates that carrying out dehydrogenation, isomerization, and ethenolysis simultaneously is a promising strategy for converting polyethylene plastic waste into propylene, a commodity with the second highest demand for use as a raw material for the manufacture of polypropylene.

Xu et al. [41] reported a relevant strategy for transforming polyethylene (**1**) and polypropylene (**2**) into fatty acids with an approximate conversion rate of 80% and average molar masses between 700 and 670 Daltons, respectively. The first stage is a temperature gradient pyrolysis that leads to the formation of waxes. The waxes are oxidized at the end of the chain with air and heating in the presence of manganese stearate, forming fatty acids (**21** and **22**), which can then be saponified to form soaps or surfactants (**23** and **24**) (Figure 2). The authors pointed out that this process can be converted to an industrial scale with economic viability.

As mentioned earlier, the pyrolysis of polyolefin waste used in plastic materials, in the absence of oxygen, can partially transform it into oils with high olefin concentrations. This oil can be a viable raw material for many new reactions. Considering that crude oil, natural gas, and naphtha have low olefin concentrations (3% by weight), these oils are relevant for fine chemicals. Within this context, Huber and collaborators prepared these pyrolysis oils with approximately 60% olefin by weight. This mixture was used to produce aldehydes through the hydroformylation reaction (Figure 3) [42]. Briefly, the hydroformylation reaction involves the addition of synthesis gas (“syngas”), a mixture of CO and H_2_, to olefins in the presence of a catalyst, leading to the formation of aldehydes at carbon 1 (**25**) or carbon 2 (**26**) [43]. This reaction is advantageous in terms of atomic economy, forming aldehydes on the terminal olefins selectively and valuable end products and intermediates in the synthesis of other chemical products such as alcohols, esters, and amines. Regioselectivity can be controlled through the choice of catalyst and reaction conditions, making hydroformylation a versatile tool in synthetic organic chemistry. This route produces high-value-added chemicals from post-consumer recycled polyethylene, which can reduce greenhouse gas emissions. The aldehydes obtained from pyrolysis oils are abundant and are synthetic platforms for the production of mono/dial alcohols by reduction or mono/dicarboxylic acids or mono/diamines by oxidative processes.

Zhang et al. [44] reported an innovative method in polyethylene cracking reactions: a one-pot low-temperature endothermic aromatization catalytic method. This method can convert polyethylene of different average molecular masses directly into liquid mixtures of valuable alkylated aromatic compounds (**29**) (Figure 4, route a). The heterogeneous catalyst used was Pt/Υ-Al_2_O_3_ (0.200 g, containing 1.5% (by weight) of Pt dispersed as ~1 nm nanoparticles) at 280 °C (±5 °C). This method uses no solvent or H_2_, and the liquid/waxy products reached 80% by mass and volatile hydrocarbons (approximately 15% by weight), demonstrating how waste polyolefins can be a viable raw material for generating aromatic hydrocarbons. Another innovative approach is the selective modification of the terminal methyl with hydrophilic groups on the carbon chains. This transformation can lead to many products for industrial applications. Zirconium-catalyzed reactions of saturated hydrocarbons with Zr(O*t*Bu)_3_ lead to the formation of organoaluminium compounds that form alcohols when exposed to air. Kanbur et al. [45] showed that the Zr(O*t*Bu)_3_@SiO_2_/Al_2_O_3_ catalyst is capable of catalytically illuminating the terminal C-H bond of polyethylene at 150 °C followed by exposure to air to provide n-dodecanol (**30**) as the main product, revealing selectivity for the activation of the methyl group without significant chain breakage, thus allowing C-H illumination of the methyl group of polyethylenes, polypropylene, polystyrene, and poly-α-olefin oils. This reaction was tested with various polyolefins and yielded an oil with 68% of chains containing alcohol functionality after the treatment of the reaction medium (Figure 4, rote b).

## 4. Cracking with Hydrogen Under Catalysis

Catalytic hydrogenolysis, a method for recycling polyolefin plastics like polyethylene and polypropylene, is gaining attention as a profitable and effective solution to the plastics crisis, with various zeolite-based catalysts showing potential for producing high-yield fuel oils similar to diesel.

In recent years, many publications have explored catalytic hydrogenolysis (pyrolysis under a hydrogen atmosphere) using various catalysts, making it the focus of large companies because it is a profitable investment and a solution to the plastics crisis [46,47]. Hydrogenolysis or hydrocracking is a reductive catalytic method for deconstructing polyolefin waste from plastics. It is a viable technology for recycling, especially for polyethylene and polypropylene. However, the issue lies in the yield of transforming these macroplastics into fuel oil with a high oil yield, preferably similar to diesel fuel (Figure 5). Zheng et al. listed various types of catalysts used in catalytic hydrogenolysis for various types of plastics. Most of the catalysts used zeolites as solid support (Ni/Co in montmorillonites, BaO, zeolite HZSM-5, Fe-modified with zeolite ZSM-5, zeolite Fe(3)-HY, zeolite, Fe/Al_2_O_3_ zeolite, ZnO zeolite, NiO/HY zeolite, SiC foam on ZSM-5, modified Mordenite, laboratory-synthesized ZSM-5 zeolite, ZAP USY zeolite, activated carbon) [48].

More recently, other types of catalysts have been studied. The most recent catalysts and their performance in hydrogenolysis to recover plastic waste are presented below.

Celik et al. [49] studied the hydrogenolysis of polyethylene with H_2_ at 170 psi and 300 °C catalyzed by Pt nanoparticles dispersed in (Pt/SrTiO_3_) without solvent. Under these conditions, polyethylene (Mn = 8000–158,000 Da) is converted into high-quality oil (Mn = 31,000 Da) that can be used as lubricants and waxes. Ruthenium is the metal used in the most active hydrogenolysis catalysts, but it is an expensive metal and produces a lot of methane. Wang et al. [50] explored the activity and economics of ruthenium-modulated tungsten zirconia (Ru-WZr) in the hydrogenolysis of low-density polyethylene and found that the deconstruction rate is fast and produces oil in the diesel range and under mild conditions (523 K and 50 bar H_2_ for 2 h). The Ru-WZr catalyst significantly suppresses methane and produces a distribution of heavier carbon products that are valuable for fuel and wax/lubricant base oil. Catalytic hydrogenolysis has the potential to convert HDPE, which comprises around 30% of plastic waste, into valuable alkanes. Most research has focused on increasing the activity of laboratory-grade HDPEs that have a low molecular weight, with a limited understanding of product distribution. No efficient catalysts are available for consumer products due to their lower reactivity. This study targets HDPE used in bottle caps, a waste product generated globally at a rate of approximately one million units per hour.

Ultrafine ruthenium particles (1 nm) supported on titania (anatase) achieved up to 80% conversion to light alkanes (C1–C45) under mild conditions (498 K, 20 bar H_2_, 4 h) and were reused for three cycles. Small ruthenium nanoparticles were critical to achieving relevant conversions as the activity decreased dramatically with particle size.

Jaydev et al. [51] also studied the catalytic hydrogenolysis of high-density polyethylene with ruthenium nanoparticles supported on titanium oxide and achieved 80% conversion to light alkanes (C1–C45) under mild conditions (498 K, 20 bar H_2_, 4 h) and with the reuse of the catalyst for three cycles. Rorrer et al. [52] studied ruthenium nanoparticles supported on carbon (Ru/C) as a heterogeneous hydrogenolysis catalyst, which proved to be highly active in converting polyethylene macroplastic waste into liquid and gaseous alkanes. They also studied the hydrogenolysis of polypropylene in the absence of solvent under mild conditions (200–250 °C, 20–50 bar of H_2_).

Zhao et al. [53] carried out catalytic hydrogenolysis of polyethylene with heterogeneous nickel-based catalysts at 280 °C and a cold hydrogen pressure of 3 MPa. The Ni/SiO_2_-supported catalyst showed the highest activity, with up to 81.18% of hydrocarbons (C4–C22). The result is comparable to catalysts using noble metals, producing iso-alkane in the C5–C32 range with conversion above 68%. Du et al. [54] carried out one-step solvent-free hydrogenolysis with skeletal rearrangements promoted by a catalyst of polyolefin plastic waste in high-value gasoline, diesel, and light lubricants with highly branched chains. The use of the bifunctional Rh/Nb_2_O_5_ catalyst takes place under mild conditions. The metallic Rh disperses in Nb_2_O_5_ (strong Brønsted acid), which breaks the long carbon chains. The β-scission of the alkylcarbenium ions increases the catalytic hydrogenolysis and isomerization of the polyolefins. This technology is economically viable and could accelerate the circular economy of plastics.

Xu et al. [55] have also shown that it is possible to obtain synthesis gas with solar irradiation with the help of water. For example, commercial plastic bags could be efficiently photoconverted into renewable synthesis gas using Co-Ga_2_O_3_ catalyst nanosheets, with hydrogen and carbon monoxide formation rates of 647.8 and 158.3 μmol∙g^−1^∙h^−1^, respectively. Water is photoreduced into hydrogen while plastics are photodegraded into carbon dioxide, which is further selectively photoreduced into carbon monoxide.

Ni_2_Al_3_-catalyzed pyrolysis of mixed polyolefin plastics for 5 to 120 min at 250–310 °C transforms these polymers into natural gas with carbon gas yields reaching 89.6% [56]. The gradual catalytic cleavage of the C-C bonds in polypropylene occurs preferentially at the terminal C-C bond in the side chain, with a low energy barrier. The gas generated was composed of 99.9% CH_4_ and 0.1% C2–C4 hydrocarbons, and the catalyst had good recyclability. This process of catalytic production of natural gas with polyolefin plastic waste is a more economically competitive process based on the current price of natural gas.

One of the challenges of fragmentation in polyolefin chains is the formation of methane (usually >20%) during hydrogenolysis. Overcoming this limitation brings economic benefits when it comes to producing liquid fuels. Chu et al. [57] solved this problem by using a Ru single-atom catalyst supported on CeO_2_ and demonstrated its efficiency in producing only 2.2% methane and a liquid fuel yield of over 94.5% at 250 °C for 6 h. This remarkable catalytic activity and selectivity of the catalyst in the hydrogenolysis of polyolefins offers various opportunities for plastic recycling. Tomer et al. [58] also studied the hydrogenolysis of polypropylene waste to form lighter liquid hydrocarbons with the Ru/CeO_2_ catalyst. The catalyst was used in the proportion of 2% by weight of Ru/CeO_2_ in the crystalline nanocube form, forming the liquid phase between 34 and 58% (220 °C, 16 h, 30 bar H_2_). Vance et al. [59] succeeded in the hydrogenolysis of polyethylene using Ni supported on SiO_2_. This reusable catalyst is very active at moderate temperatures and under H_2_ pressure (300 °C, 30 bar H_2_). Its performance is comparable to catalysts based on Ru and Pt but with the advantage of having a much lower cost. The maximum yields of liquid products were 65% by weight (n-alkanes, iso-alkanes, cyclics, aromatics, etc.). The cleavage mechanism is sensitive to the size of the chain and the breaking point of the C-C bond of the polyolefins (polyethylene, polypropylene, and polystyrene). The advantage of this procedure is its ability to hydrolyze polyolefin mixtures. Cobalt was also immobilized on SiO_2_ to produce liquid-range hydrocarbons (C5–C30) at 200–300 °C and 20–40 bar H_2_ for 2–36 h with high selectivity from polyethylene [60]. The liquid product yield was 55%, comprising 75% non-solid products, with gas yields limited to approximately 19%.

The platinum complex nanoparticle catalyst (Pt(II) acetylacetonate or trimethyl(methylcyclopentadienyl)platinum) supported by the chemical insertion of organometallics on the surface of SrTiO_3_ nanocuboids was able to promote the hydrogenolysis of polypropylene to provide liquid products with a narrow range of molecular dispersion [61]. These catalysts were obtained by calcination and had Pt nanoparticles of 1.0–1.5 nm deposited on the SrTiO_3_ nanocuboids. These catalysts hydrogenolyzed polypropylene into liquid products with >95% yield and average molecular weights of 200–300 Da at 300 °C and 180 psi H_2_.

Many inorganic materials or waste can be used as catalysts. Red mud is a by-product generated during the Bayer process of extracting alumina (aluminum oxide) from bauxite. Rahman et al. used this material, after drying, as a catalyst to increase the productivity of the hydrogenolysis conversion of various plastic materials. The results indicated that the conversion to oil ranged from 14 to 80% (*w*/*w*) depending on the type of polymer [62].

## 5. Perspectives and Challenges to Plastic Depolymerization

Current challenges in the pyrolysis and hydrogenolysis of macroplastic polyolefins include feedstock variability; mixed plastic types and contaminants that hinder consistent processing; scalability; energy efficiency; product quality; environmental concerns; waste byproducts; and poor management. Future research should focus on developing efficient catalysts, integrating renewable energy sources, improving process standardization, and enhancing circularity through optimized chemical recycling pathways.

The pyrolysis method appears to be more versatile as it operates at higher temperatures compared to hydrogenolysis, which functions under milder conditions but requires high-pressure hydrogen. Additionally, pyrolysis produces a broad spectrum of products, making it suitable for processing mixed and contaminated plastics. In contrast, hydrogenolysis provides more targeted outputs, often favoring high-value chemicals. Furthermore, the reactor design for pyrolysis is simpler, which can make the process more accessible and cost-effective in certain applications.

## 6. Conclusions

Some end-of-life macroplastic waste can be recycled through mechanical processes, but this accounts for only about 25% of all macroplastics. The recovery of plastics through chemical recycling into chemicals, fuels, lubricants, and paraffin waxes is still under investigation. The two most studied technologies for this process are catalytic thermal pyrolysis and catalytic hydrogenolysis. Both have been extensively researched in recent years to determine the best reaction conditions, applicability to various types of polymers, and the most economical and efficient catalysts. These technologies also offer prospects for producing fine chemical compounds, such as acids, alcohols, and aldehydes. These are crucial steps, but without a reduction in the production of plastics and improvement in the selective collection of macroplastics, no progress will be made in addressing the global plastics crisis.

Finally, while our manuscript focuses on the processes of pyrolysis and hydrogenolysis, we acknowledge that both methods, especially if not optimized, can result in associated carbon emissions. However, we believe that these processes, when integrated with renewable energy sources or advanced carbon capture technologies, can significantly reduce their environmental impact.

## Data Availability

The data are contained within the article.

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
