# Peer review of "Recent Progress in Polyolefin Plastic: Polyethylene and Polypropylene Transformation and Depolymerization Techniques"

_molecules, 2024, doi:10.3390/molecules30010087_

Round 1
Reviewer 1 Report
Comments and Suggestions for Authors
I appreciate how the authors described the "chemolysis" methods, but the text does not mention the necessity of macroplastic waste pretreatment. In my opinion, it should be at least mentioned as well as the sorting of polymers within the polyolefins group and the impact of nonsorting for the listed methods, because it could be crucial for the outputs.
Another possible technique is e.g. gasification, why it is not mentioned?
The authors swapped the PET and PVC formulas on page 2 of the text.
On line 68 the verb "includes" should be used in plural, and in the text in the description of Scheme 4, there should be "u" in the word "route".
Still, I'm curious if it's possible to compare the mentioned methods from any point of view (polymer, reactors, temperatures, outputs...) because that would be a brilliant conclusion to the review.
Author Response
Comments 1: I appreciate how the authors described the "chemolysis" methods, but the text does not mention the necessity of macroplastic waste pretreatment. In my opinion, it should be at least mentioned as well as the sorting of polymers within the polyolefins group and the impact of nonsorting for the listed methods, because it could be crucial for the outputs.
Response 1: We thank the reviewer for pointing out the absence of this information. We rewrite the text in yellow. “This article focuses exclusively on chemical upgrading processes, specifically chemolysis by pyrolysis and hydrogenolysis. It does not address the biochemical degradation of plastics or the production of biodegradable polymers. It is important to note that chemolysis refers to the chemical breakdown of polymers, a process commonly used to recycle plastics into their monomers or other valuable compounds. To ensure the efficiency of chemolysis and the production of high-quality outputs, waste pretreatment is essential. This involves steps such as shredding, washing, and separating contaminants”.
Comments 2: Another possible technique is e.g. gasification, why it is not mentioned?
Response 2: We appreciate the comment raised and fully agree with your observation. Gasification is a process that converts plastic waste into a mixture of gases, primarily syngas (a blend of carbon monoxide (CO) and hydrogen (Hâ‚‚)), by heating plastics at high temperatures (typically 700–1200°C) in the presence of a controlled amount of oxygen, air, or steam. While we recognize that pyrolysis and hydrogenolysis have been the most extensively researched technologies in recent years—hence our focus on them—we have now introduced a paragraph to highlight the significance of gasification as another transformative process for plastic waste management.
Comments 3: The authors swapped the PET and PVC formulas on page 2 of the text.
Response 3: We agree with this observation. Modifications have been made in the text.
Comments 4: On line 68 the verb "includes" should be used in plural, and in the text in the description of Scheme 4, there should be "u" in the word "route".
Response 4: We agree with this observation. Modifications have been made in the text
Comments 5: Still, I'm curious if it's possible to compare the mentioned methods from any point of view (polymer, reactors, temperatures, outputs...) because that would be a brilliant conclusion to the review.
Response 5: We agree with this observation. “The pyrolysis method appears to be more versatile, as it operates at higher temperatures compared to hydrogenolysis, which functions under milder conditions but requires high-pressure hydrogen. Additionally, pyrolysis produces a broad spectrum of products, making it suitable for processing mixed and contaminated plastics. In contrast, hydrogenolysis provides more targeted outputs, often favoring high-value chemicals. Furthermore, the reactor design for pyrolysis is simpler, which can make the process more accessible and cost-effective in certain applications”. This text was included in the conclusion.
Kindly,
Acácio Silva de Souza
Vitor Francisco Ferreira
Reviewer 2 Report
Comments and Suggestions for Authors
Manuscript ID: molecules-3316088
Title: Recent Progress in Plastic Transformation and Depolymerization Techniques
The manuscript is devoted to the global problem of combating the accumulation of plastic waste in the biosphere, which is considered by the authors as a resource for recycling using modern pyrolysis and hydrogenolysis technologies as the latest alternative to disposal. Today, the share of plastic waste subject to recycling is extremely small compared to the increasing production volumes. Therefore, the problem of recycling plastic waste and the use of effective and environmentally friendly methods for this is relevant.
The manuscript is a review and trends in optimizing pyrolysis and hydrogenolysis methods currently being developed for recycling plastic waste. When reading the manuscript, the reviewer had questions that need clarification and motivation.
In the section "INTRODUCTION" it seems appropriate to specify and provide current quantitative indicators of the volumes of production of the main synthetic plastics from oil, as well as accurate data on the volumes of burial, incineration and recycling of plastic waste. Global environmental damage and problems associated with the accumulation of plastic waste in the biosphere are not motivated. Microplastic, which has objectively turned into a bugbear and is today considered as a key threat to biota and human health, the authors consider exclusively as a resource for obtaining products with added value, and this is not fair.
The authors consider two processes for recycling plastic waste - pyrolysis and hydrogenolysis, implemented in various versions - with varying temperatures, with the use of various catalysts, oxygen or hydrogen and without them, etc. In this area, the researcher is not a deep specialist, so he cannot assess the completeness and value of the presented review of modern publications and their discussions.
It is known that the fight against plastic waste to a certain, but not global!! extent can be solved and is partially solved today by recycling plastic waste. However, this requires significant labor and energy costs, since the following actions are necessary for this: collecting and sorting plastic waste, separating the collected waste by type of plastic, washing, drying, grinding and only then processing into new products, materials and polymer products. As is known, the possible ways of reducing the giant waste of synthetic plastics considered in modern literature are recycling, which can be divided into a number of main areas: incineration, pyrolysis, recycling and recycling, as well as burial - this is a time bomb for future generations, which is today a global means of "fighting" plastic waste. This is accompanied by the growth of landfills and the occupation of fertile and arable lands for them.
The authors passed over in silence the fact that burning and pyrolysis of plastic waste do not improve the environmental situation. Moreover, high-temperature processes are expensive processes, and they also lead to the formation of highly toxic and supertoxic compounds, including furans and dioxins. The need for a gradual transition to biodegradable polymeric materials that fit into global biospheric cycles without the formation of toxic decay products, i.e., harmless to the environment and biota, is recognized as a radical solution to the problem of clogging the biosphere with plastic and especially microplastics. It is advisable to provide the available data that greenhouse gas emissions during the recycling and processing of synthetic oil-based plastics are 1.5-1.7 times higher compared to biodegradable polymeric materials (polylactides, polyhydroxyalkanoates, etc.) - an important comment on the need to reduce the carbon footprint in the biosphere, which the authors also do not mention when discussing the processes of pyrolysis and hydrogenolysis for the recycling of plastic waste, which is the subject of the manuscript
Conclusion: The reviewer is not an expert in the field of pyrolysis and hydrogenolysis technologies of polyolefin waste, therefore, he cannot objectively assess the completeness and significance of the presented review of published materials. However, the focus of the work on the need to attract giant plastic waste for the purpose of their elimination through recycling is not justified given modern concepts and trends. This should be replenished in the manuscript during its processing.
Author Response
Comments 1: In the section "INTRODUCTION" it seems appropriate to specify and provide current quantitative indicators of the volumes of production of the main synthetic plastics from oil, as well as accurate data on the volumes of burial, incineration and recycling of plastic waste. Global environmental damage and problems associated with the accumulation of plastic waste in the biosphere are not motivated.
Response 1: We thank the reviewer for the observation. We introduced the text: “Worldwide, more than 330 million tons of plastic are produced annually, with a considerable increase in production during the COVID-19 pandemic period from2019-2021. It is estimated that there are already 4.9 billion tons of plastic waste of varying sizes and chemical compositions, ubiquitous to all natural habitats, and those materials are spread in terrestrial and aquatic ecosystems. Projections indicate that in 2050 this amount should increase by 12 billion metric tons. [2”]
Comments 2: Microplastic, which has objectively turned into a bugbear and is today considered as a key threat to biota and human health, the authors consider exclusively as a resource for obtaining products with added value, and this is not fair.
Response 2: We agree with the reviewer, but the problem with microplastics and nanoplastics (MNPs) is that they are difficult to collect in the environment. These particles have been distributed worldwide for a long time, but they are actually being considered as emerging pollutants and their potential risks to health have been assessed. These particles are an emerging global environmental contaminant that affects living beings and ecosystems, however, little is known about the effects to MNPs exposure and absorption by the human body. If macroplastics are collected properly, recycled or used as raw materials for other chemicals, the amount of MNPs will decrease, but the problem of their collection will persist for a long time.
Comments 3: The authors consider two processes for recycling plastic waste - pyrolysis and hydrogenolysis, implemented in various versions - with varying temperatures, with the use of various catalysts, oxygen or hydrogen and without them, etc. In this area, the researcher is not a deep specialist, so he cannot assess the completeness and value of the presented review of modern publications and their discussions.
Response 3: We aim to highlight plastic recycling technologies, specifically pyrolysis and hydrogenolysis, in response to the significant amount of microplastic pollution in the environment. Modern reviews offer a comprehensive comparison of these methods, detailing their advantages, limitations, and optimal applications. This information is invaluable for researchers, policymakers, and industry professionals in selecting the most suitable technology for specific recycling objectives, particularly in the context of plastic waste management.
Comments 4: It is known that the fight against plastic waste to a certain, but not global!! extent can be solved and is partially solved today by recycling plastic waste. However, this requires significant labor and energy costs, since the following actions are necessary for this: collecting and sorting plastic waste, separating the collected waste by type of plastic, washing, drying, grinding and only then processing into new products, materials and polymer products. As is known, the possible ways of reducing the giant waste of synthetic plastics considered in modern literature are recycling, which can be divided into a number of main areas: incineration, pyrolysis, recycling and recycling, as well as burial - this is a time bomb for future generations, which is today a global means of "fighting" plastic waste. This is accompanied by the growth of landfills and the occupation of fertile and arable lands for them.
Response 4: We partly agree with the reviewer’s comments. However, we would like to emphasize that the issue of macroplastics is already well-established, with billions of tons of plastic dispersed throughout the environment. Oil companies are largely unwilling to discuss reducing the raw materials used in plastic production. While recycling is undoubtedly a key solution, it is limited in effectiveness, as it can only be performed a few times. Currently, 37.3% of macroplastics are incinerated, 37% are discarded, and only 25.7% are recycled. Recycling remains an important option, but reducing plastic production and consumption may prove to be more viable in the long run.
Comments 5: The authors passed over in silence the fact that burning and pyrolysis of plastic waste do not improve the environmental situation. Moreover, high-temperature processes are expensive processes, and they also lead to the formation of highly toxic and supertoxic compounds, including furans and dioxins. The need for a gradual transition to biodegradable polymeric materials that fit into global biospheric cycles without the formation of toxic decay products, i.e., harmless to the environment and biota, is recognized as a radical solution to the problem of clogging the biosphere with plastic and especially microplastics. It is advisable to provide the available data that greenhouse gas emissions during the recycling and processing of synthetic oil-based plastics are 1.5-1.7 times higher compared to biodegradable polymeric materials (polylactides, polyhydroxyalkanoates, etc.) - an important comment on the need to reduce the carbon footprint in the biosphere, which the authors also do not mention when discussing the processes of pyrolysis and hydrogenolysis for the recycling of plastic waste, which is the subject of the manuscript
Response 5: Thank you for the insightful comment. We agree that the reduction of the carbon footprint in the biosphere is a critical aspect, particularly when discussing the recycling of plastic waste. Undoubtedly, recycling is a way to reduce the entry of new polymers into the environment. However, chemical recycling is also attractive because it generates higher value-added products through various types of chemical conversions and offsets the energy cost. With the development of new catalysts, this process will probably be more advantageous with the formation of a smaller amount of by-products. Currently, chemical recycling is limited to condensation polymers and requires large volumes of plastics to be profitable. Among the applicable methods, depolymerization, a process that began a few years ago, is gaining more and more prominence in the use of macroplastics, with the search for new procedures and new catalysts being expanded due to the environmental pollution crisis by macro and microplastics.While our manuscript focuses on the processes of pyrolysis and hydrogenolysis, we acknowledge that both methods, especially if not optimized, can have associated carbon emissions. However, we believe that these processes, when integrated with renewable energy sources or advanced carbon capture technologies, can significantly reduce their environmental impact. We added this coment on the conclusion.
Comments 6: However, the focus of the work on the need to attract giant plastic waste for the purpose of their elimination through recycling is not justified given modern concepts and trends. This should be replenished in the manuscript during its processing.
Response 6: Thank you for your comment. We understand your point regarding the focus on large-scale plastic waste elimination through recycling. While the manuscript primarily emphasizes alternative recycling processes such as pyrolysis and hydrogenolysis, we acknowledge the importance of broader context, particularly with respect to modern trends in waste management and plastic recycling. The review updates the discussion on current approaches and trends in plastic waste management, particularly focusing on the limitations of conventional recycling methods and the increasing importance of alternative solutions like chemical recycling. This will help justify the focus on pyrolysis and hydrogenolysis as part of a broader strategy to address the global plastic waste crisis.
Kindly,
Acácio Silva de Souza
Vitor Francisco Ferreira
Reviewer 3 Report
Comments and Suggestions for Authors
In the manuscript titled “Recent Progress in Plastic Transformation and Depolymerization Techniques”, the authors summarized recent developments in the deconstruction of plastics, with a special focus on polyolefins, highlighting two primary methods for plastics deconstruction: pyrolysis and hydrogenolysis. Pyrolysis is a thermal decomposition process in which polyolefins are broken down at elevated temperatures in the absence of oxygen, yielding a range of valuable products, such as bio-oils, gases, etc.. Hydrogenolysis, on the other hand, involves the use of hydrogen to cleave chemical bonds in the polymer chains resulting in the formation of smaller hydrocarbons. Although the authors provided numerous examples, yet the manuscript lacks depth and clarity. The content of the manuscript remains largely superficial, and the writing fails to highlight the essential points. The reviewer recommends the following revisions to the manuscript before it can be accepted for publication:
1. The structure and writing of the ‘abstract’ and ‘introduction’ sections does not reflect advanced understanding of the subject. With a more refined approach to the content and writing, the authors could better capture the attention of the readers and emphasize the crucial aspects of the different recycling strategies discussed in the manuscript.
2. It would be beneficial to the readers if the authors could highlight the advantages of pyrolysis and hydrogenolysis in comparison to other recycling methods.
3. The authors mentioned in the title ‘Plastic transformation and depolymerization’, yet they focused only on polyolefins in the manuscript. For a more comprehensive review, it would be beneficial to include additional polymers such as polyurethanes, polyamides, polystyrene, PET, polyvinyl chloride, polylactic acid, etc. These polymers are used in various industries and present unique challenges for depolymerization through processes like pyrolysis and hydrogenation. By discussing the depolymerization strategies of these materials, the review would offer a deeper understanding of the recycling strategies across a broader range of polymers used in the industries.
4. Figures (2 – 4): It would be beneficial to improve the design and labeling of the figures to enhance the overall visual appeal and understanding of the data.
5. Check for typos, e.g., recyclin, hydrogenolisis, and pyrolisis in Fig. 3, polyethilene in Scheme 4, CH4 on Pg 13 line 374, etc..
Author Response
Comments 1: The structure and writing of the ‘abstract’ and ‘introduction’ sections does not reflect advanced understanding of the subject. With a more refined approach to the content and writing, the authors could better capture the attention of the readers and emphasize the crucial aspects of the different recycling strategies discussed in the manuscript.
Response 1: Thank you for your insightful comment regarding the structure of our manuscript, Recent Progress in Plastic Transformation and Depolymerization Techniques. We understand the critical role an abstract plays in summarizing and framing the importance of a scientific study. An abstract is not merely a condensed version of the manuscript but also serves as a gateway for readers to grasp the scope, significance, and key findings of the research at a glance. We believe this enhancement will significantly improve the manuscript's accessibility and appeal, ensuring that its message is both clear and impactful. Thank you for your suggestion, which has guided us in making this important improvement.
Comments 2: It would be beneficial to the readers if the authors could highlight the advantages of pyrolysis and hydrogenolysis in comparison to other recycling methods.
Response 2: A paragraph has been added to the conclusion comparing pyrolysis and hydrogenolysis across three key aspects.
Comments 3: The authors mentioned in the title ‘Plastic transformation and depolymerization’, yet they focused only on polyolefins in the manuscript. For a more comprehensive review, it would be beneficial to include additional polymers such as polyurethanes, polyamides, polystyrene, PET, polyvinyl chloride, polylactic acid, etc.
Response 3: The choice of scope in a scientific study is a critical decision that impacts both the depth and clarity of the work. While the reviewer’s suggestion to broaden the analysis to include a wider range of polymers is valuable, the authors’ decision to concentrate on polyethylene (PE) and polypropylene (PP) is well-justified. These polymers dominate global plastic production and are significant contributors to the microplastic crisis, making them pivotal to any discussion on environmental pollution. Broadening the scope to include all polymer types would indeed provide a more comprehensive overview, but it risks diluting the focus on these two critical contributors. By narrowing the lens, the study can delve deeper into the specific challenges and solutions related to PE and PP, offering actionable insights that may have the greatest environmental impact.
Comments 4: Figures (2 – 4): It would be beneficial to improve the design and labeling of the figures to enhance the overall visual appeal and understanding of the data.
Response 4: Figure 2 has been revised to clearly illustrate that the circle sizes reflect the proportional data. Figure 3, which outlines strategies for the recovery of macroplastics, is presented in a clear and detailed manner. Figure 4 acts as a comprehensive summary, focusing on the potential outcomes of pyrolysis and hydrogenolysis processes for plastics. In our view, these figures effectively convey the intended information, and only minimal adjustments may be needed to ensure they remain concise and explicit.
Comments 5: Check for typos, e.g., recyclin, hydrogenolisis, and pyrolisis in Fig. 3, polyethilene in Scheme 4, CH4 on Pg 13 line 374, etc…
Response 5: We thank the reviewer for the observation. The correction has been made accordingly.
Comments 6: In the first paragraph on page 3, several detail errors are present, such as “3a-3l”, “3i”, and “3l”.
Response 6: We agree with this observation. The modifications have been made in the text.
Comments 7: In Scheme 1, only A, B, and C are present, and there is no Figure D. The authors are advised to carefully review this.
Response 7: We thank the reviewer for the perception. The change has been made accordingly in the text highlighted in yellow.
Comments 8: In Table 1, the “Room temperature” entry in entry 12 is incorrect.
Response 8: We thank the reviewer for the observation. The correction has been made accordingly.
Kindly,
Acácio Silva de Souza
Vitor Francisco Ferreira
Round 2
Reviewer 2 Report
Comments and Suggestions for Authors
The paper can be accepted in present form
Author Response
Comments 1: The paper can be accepted in present form.
Response 1: Thank you so much for reviewing this manuscript. Thank you for your favorable opinion to accept this paper.
Reviewer 3 Report
Comments and Suggestions for Authors
In the revised manuscript titled “Recent Progress in Plastic Transformation and Depolymerization Techniques”, the authors have addressed some of the suggestions. However, the reviewer feels that the manuscript has not been improved enough and is not suitable for publication in the journal.
Author Response
Comments 1: In the revised manuscript titled “Recent Progress in Plastic Transformation and Depolymerization Techniques”, the authors have addressed some of the suggestions. However, the reviewer feels that the manuscript has not been improved enough and is not suitable for publication in the journal.
Response 1: Thank you so much for reviewing this manuscript and for your contributions and suggestions. Our research group will strive to improve our knowledge on the subject to contribute more effectively to this important and current area of science.